# EVALUATING GANS VIA DUALITY

## ABSTRACT

Generative Adversarial Networks (GANs) have shown great results in accurately modeling complex distributions, but their training is known to be difficult due to instabilities caused by a challenging minimax optimization problem. This is especially troublesome given the lack of an evaluation metric that can reliably detect non-convergent behaviors. We leverage the notion of duality gap from game theory in order to propose a novel convergence metric for GANs that has low computational cost. We verify the validity of the proposed metric for various test scenarios commonly used in the literature.

## 1 INTRODUCTION

In the past few years, generative models have become extremely popular in the machine learning community. This is largely due to the recent advances in the field of deep learning, which allowed deep neural generators to produce remarkable results for various tasks, including for example image generation (Radford et al., 2015). Two notable approaches in this area are variational auto-encoders (VAEs) (Kingma & Welling, 2013; Rezende et al., 2014), and generative adversarial networks (GAN) (Goodfellow et al., 2014). In this paper, we focus on GANs, which are especially attractive as they circumvent the notoriously hard optimization of the data likelihood and instead use an adversarial game approach for training a generator.

Let us denote the data distribution by $p_{\text{data}}(\mathbf{x})$ and the model distribution by $p_{\mathbf{u}}(\mathbf{x})$. A probabilistic discriminator is denoted by $D_{\mathbf{v}} : \mathbf{x} \mapsto [0; 1]$ and a generator by $G_{\mathbf{u}} : \mathbf{z} \mapsto \mathbf{x}$. The GAN objective is:

$$\min_{\mathbf{u}} \max_{\mathbf{v}} M(\mathbf{u}, \mathbf{v}) = \frac{1}{2}\mathbb{E}_{\mathbf{x} \sim p_{\text{data}}} \log D_{\mathbf{v}}(\mathbf{x}) + \frac{1}{2}\mathbb{E}_{\mathbf{z} \sim p_{\mathbf{z}}} \log(1 - D_{\mathbf{v}}(G_{\mathbf{u}}(\mathbf{z}))) . \tag{1}$$

Each of the two players (generator/discriminator) tries to optimize their own objective, which is exactly balanced by the loss of the other player, thus yielding a two-player zero-sum minimax game. Standard GAN approaches aim at finding a pure Nash Equilibrium by using traditional gradient-based techniques to minimize each players cost in an alternating fashion. However, the minimax objective of GANs makes the optimization process challenging. One of the central open issues is the non-convergence problem, which in practice leads to oscillations between different kinds of generated samples (Metz et al., 2016). Many different techniques have been proposed to address the shortcomings of the original technique introduced by Goodfellow et al. (2014).

Given the plethora of GAN-like approaches as well as different training methods, the community is now facing the problem of determining which approaches produce "better" generative models. This is in itself a challenging question as there is no clear evaluation criterion. In a recent large-scale empirical study, Lucic et al. (2018) showed that using various evaluation measures, many state-of-the-art models are able to reach similar scores with enough tuning. In a recent survey, Borji (2018) discussed the merits of various evaluation metrics, pointing out that there is no clear consensus regarding which metric is the most appropriate to evaluate the quality of a generative model. While many metrics achieve reasonable discriminability (i.e. ability to distinguish generated samples from real ones), they also tend to have a high computational complexity. Many existing metrics are also specific to datasets of natural images, but we are also starting to notice the success of GANs in other fields such as in cosmology (Rodriguez et al., 2018) or in the medical domain (Schlegl et al., 2017).

Another problem related to the evaluation of samples produced by a GAN is the lack of evaluation procedures to detect convergence. While various approaches have analyzed the convergence properties of alternating gradient-based techniques, most of them require the objective function to

be convex-concave (Goodfellow et al., 2014; Nowozin et al., 2016) with the exception of Grnarova et al. (2018) who prove convergence for semi-concave zero-sum games. In general, alternating gradient descent can fail to converge (Salimans et al., 2016). While this non-convergence behavior can in practice be visually recognized for some low-dimensional examples (such as a 2D mixture of Gaussian), this is in general more difficult in high-dimensional spaces due to the lack of a convergence metric. This has been pointed out before (see e.g. Mescheder et al. (2018) and (Fedus et al., 2018) who examine convergence on 2D problems and problems for which the true data distribution is known.

One of the common challenges that practitioners are facing is when to stop training. In particular, it is well known that the curves of the discriminator and generator losses oscillate (see Figures 12 and 13) and are non-informative as to whether the model is improving or not (Arjovsky et al., 2017). This is especially troublesome when a GAN is trained on non-image data in which case one might *not* be able to use visual inspection or FID/Inception score as a proxy.

In this paper, our main contribution is to leverage existing ideas from game theory to propose a novel and computationally efficient convergence metric for GANs. The principle we follow is based on the theory of duality that was developed after Von Neumann (1928) derived the minimax theorem for zero-sum games. From duality, it follows that the minimax and maximin [1] are dual to each other and strong duality holds for feasible problems. This duality also implicitly gives us a way to evaluate the deviation between minimax and maximin. As an example, let's consider a two-player game between the discriminator D and the generator G. The duality gap measures the difference between the values of the game under two scenarios: when D first commits to a strategy, and then G gets to respond; and when G commits first, and then D gets to respond. In this paper, we advocate the use of the duality gap as a measure of convergence for GANs and demonstrate its effectiveness for various settings commonly considered in the literature.

As a second contribution, we also demonstrate how to use the minimax loss value in order to detect mode collapse and measure sample quality. Unlike FID or the Inception score that require labelled data or a domain dependent classifier, our metric is domain independent and does not require labels. Our experiments demonstrate that the duality gap and minimax loss are valuable tools to measure convergence and correlate well with existing evaluation metrics.

## 2   RELATED WORK

Despite the impressive empirical performance achieved by the latest GAN models (Karras et al., 2017), they are still subject to many unanswered questions. Among them is the issue of a fair evaluation procedure. Since the log-likelihood of the data is a common objective function to train a generative model, it would appear to be a sensible metric for GANs. However, its computation is often intractable and Theis et al. (2015) also demonstrate that it has severe limitations as it might yield low visual quality samples despite of a high likelihood. Perhaps the most popular evaluation metric for GANs is the inception score introduced by Salimans et al. (2016) that measures both diversity of the generated samples and discriminability. While diversity is measured as the entropy of the output distribution, the discriminability aspect requires a pretrained neural network to assign high scores to images close to training images. Various modifications of the inception score have been suggested. Gurumurthy et al. (2017) adds a term in the cost function to take into account diversity within samples in a particular class. Heusel et al. (2017) uses features from a hidden layer of the Inception Net, which are modelled as two multivariate Gaussians for the generated and true data. They then suggest using the Frechet Inception Distance (FID) between these two Gaussians to assess the quality of the samples. The authors demonstrate improvements over the inception score, especially more robustness to noise, as well as being able to detect intra-class mode dropping. Furthermore, they also demonstrate high degrees of consistency with human judgments. Although this measure appears to have nice properties from an empirical point of view, the Gaussian assumption might not hold in practice. Furthermore, FID requires labelled data in order to train a classifier. Without labels, transfer learning is possible to datasets under limited conditions (i.e. the source and target distributions should not be too dissimilar).

---

[1] In game theory, the maximin value of a player is the highest value that the player is ensured to receive without knowing the actions of the other player(s) while the minimax value of a player is the smallest value that the other player(s) can force the player to receive, without knowing the player's actions.

Another popular metric introduced by Gretton et al. (2012) is the Maximum Mean Discrepancy (MMD) which measures the dissimilarity between $p_{\text{data}}$ and $p_{\mathbf{u}}$ using independently drawn samples. It is a specific instance of an integral probability metric. A potential hurdle for MMD is that its computational complexity is quadratic in the sample size, although some linear approximation do exist (Gretton et al., 2012). Finally, it also depends on the choice of the kernel.

There are many other metrics that have been suggested over the past few years. We refer the reader to Borji (2018) for a detailed survey. Perhaps the two approaches that are the most relevant to our approach are inspired by skill rating systems in games and were both introduced in Olsson et al. (2018). The first method named tournament win rate considers a single model playing against past and future versions of itself. This measure is designed to monitor the progress of a single model as it learns during training. The second method named skill rating measures the aptitude of two different fully trained models. In some way, skill rating and tournament win rate can be seen as an approximation of the worst minimax value we advocate in this paper, where instead of doing a full-on optimization in order to find the best adversary for the fixed generator, the search space is limited to discriminators that are snapshots from training, or discriminators trained with different seeds.

We would like to conclude our discussion of related work by pointing out to the vast literature on duality used in the optimization community as a convergence criterion for min-max saddle point problem, see e.g. Nemirovski et al. (2009); Komodakis & Pesquet (2015). Some recent work by Chen et al. (2018) also used duality in order to derive a Lagrangian objective to train GANs. Although we also make use of duality, there are some major differences that are worth pointing out. Unlike prior work, our contribution does not relate to optimising GANs, but instead in showing that the duality gap can be empirically used as a proxy to track convergence. We also demonstrate it can be computed at low cost.

## 3 DUALITY GAP AS A NATURAL PERFORMANCE MEASURE

Standard learning tasks are often described as (stochastic) optimization problems; this applies to common Deep Learning scenarios as well as to classical tasks such as logistic and linear regression. This formulation gives rise to a natural performance measure, namely the test loss[2]. In contrast, GANs are formulated as (stochastic) zero-sum games. Unfortunately, this fundamentally different formulation does not allow us to use the same performance metric. In this section, we describe a performance measure for GANs, which naturally arises from a game theoretic perspective.

We start this section with a brief overview of zero-sum games, including a description of the *Duality gap* metric and some of its properties. Then we illustrate the usefulness of this metric by analyzing the idealized setting where both $G$ and $D$ have unbounded capacity (this setting was previously discussed in Goodfellow et al. (2014)).

A zero-sum game is defined by two players $\mathcal{P}_1$ and $\mathcal{P}_2$ who choose a decision from their respective decision sets $\mathcal{K}_1$ and $\mathcal{K}_2$. A game objective $M : \mathcal{K}_1 \times \mathcal{K}_2 \mapsto \mathbb{R}$, sets the utilities of the players. Concretely, upon choosing a pure strategy $(\mathbf{u}, \mathbf{v}) \in \mathcal{K}_1 \times \mathcal{K}_2$ the utility of $\mathcal{P}_1$ is $-M(\mathbf{u}, \mathbf{v})$, while the utility of $\mathcal{P}_2$ is $M(\mathbf{u}, \mathbf{v})$. The goal of either $\mathcal{P}_1/\mathcal{P}_2$ is to maximize their worst case utilities; thus,

$$\min_{\mathbf{u}\in\mathcal{K}_1} \max_{\mathbf{v}\in\mathcal{K}_2} M(\mathbf{u}, \mathbf{v}) \quad \textbf{(Goal of } \mathcal{P}_1\textbf{)}, \quad \& \quad \max_{\mathbf{v}\in\mathcal{K}_2} \min_{\mathbf{u}\in\mathcal{K}_1} M(\mathbf{u}, \mathbf{v}) \quad \textbf{(Goal of } \mathcal{P}_2\textbf{)} \quad (2)$$

The above formulation raises the question of whether there exists a solution $(\mathbf{u}^*, \mathbf{v}^*)$ to which both players may jointly converge. The latter only occurs if there exists $(\mathbf{u}^*, \mathbf{v}^*)$ such that neither $\mathcal{P}_1$ nor $\mathcal{P}_2$ may increase their utility by unilateral deviation. Such a solution is called a *pure equilibrium*, and is formally defined as follows,

$$\max_{\mathbf{v}\in\mathcal{K}_2} M(\mathbf{u}^*, \mathbf{v}) = \min_{\mathbf{u}\in\mathcal{K}_1} M(\mathbf{u}, \mathbf{v}^*) \quad \textbf{(Pure Equilibrium).}$$

While a pure equilibrium does not always exist, the seminal work of Nash et al. (1950) shows that an extended notion of equilibrium always does. Specifically, there always exists a distribution $\mathcal{D}_1$ over elements of $\mathcal{K}_1$, and a distribution $\mathcal{D}_2$ over elements of $\mathcal{K}_2$, such that the following holds,

$$\max_{\mathbf{v}\in\mathcal{K}_2} \mathbb{E}_{\mathbf{u}\sim\mathcal{D}_1} M(\mathbf{u}, \mathbf{v}) = \min_{\mathbf{u}\in\mathcal{K}_1} \mathbb{E}_{\mathbf{v}\sim\mathcal{D}_2} M(\mathbf{u}, \mathbf{v}) \quad \textbf{(Mixed Nash Equilibrium).}$$

---

[2]For classification tasks using the zero-one test error is also very natural. Nevertheless, in regression tasks the test loss is often the only reasonable performance measure.

Such a solution is called a *Mixed Nash Equilibrium (MNE)*. This notion of equilibrium gives rise to the following natural performance measure of a given pure/mixed strategy.

**Definition 1** (Duality Gap). *Let $\mathcal{D}_1$ and $\mathcal{D}_2$ be fixed distributions over elements from $\mathcal{K}_1$ and $\mathcal{K}_2$ respectively. Then the duality gap of $(\mathcal{D}_1, \mathcal{D}_2)$ is defined as follows,*

$$\text{DualGap} := \max_{\mathbf{v} \in \mathcal{K}_2} \mathbb{E}_{\mathbf{u} \sim \mathcal{D}_1} \text{M}(\mathbf{u}, \mathbf{v}) \; - \; \min_{\mathbf{u} \in \mathcal{K}_1} \mathbb{E}_{\mathbf{v} \sim \mathcal{D}_2} \text{M}(\mathbf{u}, \mathbf{v}) \, . \tag{3}$$

*Particularly, for a given* pure *strategy* $(\mathbf{u}, \mathbf{v}) \in \mathcal{K}_1 \times \mathcal{K}_2$ *we define,*

$$\text{DualGap} := \max_{\mathbf{v} \in \mathcal{K}_2} \text{M}(\mathbf{u}, \mathbf{v}) \; - \; \min_{\mathbf{u} \in \mathcal{K}_1} \text{M}(\mathbf{u}, \mathbf{v}) \, . \tag{4}$$

A well-known, straightforward property of the duality gap is that it is always non-negative. Moreover, by definition, the gap is exactly zero in (mixed) Nash Equilibrium solutions. This property is very appealing from a practical point of view, since it means that the duality gap gives us an immediate handle for measuring convergence.

Next we illustrate the usefulness of the duality gap metric by analyzing the ideal case where both the generator and discriminator have unbounded capacity. The latter basically means that the generator can represent any distribution, and the discriminator can represent any decision rule. The next proposition shows that in this case, as long as $G$ is not equal to the true distribution then the duality gap is always positive. In particular we show that duality gap is at least as large as the Jensen-Shannon divergence between true and fake distributions (which is always non-negative). We also show that if $G$ outputs the true distribution, then there exists a discriminator such that the duality gap is zero.

**Proposition 1.** *Consider the GAN game objective appearing in Equation (1), and assume that the generator and discriminator networks have unbounded capacity. Also, let $(G, D)$ be a fixed solution. Then the duality gap of $(G, D)$ is larger than the Jensen-Shannon divergence between the true distribution and the fake distribution generated by $G$. Moreover, if $G$ outputs the true distribution, then there exists a discriminator $D$ such that the Duality gap of $(G, D)$ is zero.*

## 4 Estimating the Duality Gap metric for GANs

In this section we address several aspects of estimating the duality gap metric for GANs. First we discuss the appropriate way to estimate the metric using samples. We follow with advocating another metric that enables evaluation of a generator not necessarily trained by a GAN. Finally, we describe a method for an efficient and practical computation of the duality gap.

**Appropriately estimating the duality gap from samples:** Standard supervised learning problems are often formulated as stochastic optimization programs. Thus, we do not have direct access to the expected loss, but can instead estimate it through samples. In this case, it is well known that one should split the data into training and test sets [3]. The training set is used to find a solution whose quality of the solution is estimated using a separate test set (which provides an unbiased estimate of the true expected loss).

Similarly, GANs are formulated as stochastic zero-sum games (Equation (1)). Nevertheless, in GANs the issue of evaluating the duality gap metric is more delicate. This is because we have three phases in the evaluation: **(i)** training a model $(\mathbf{u}, \mathbf{v})$, **(ii)** finding the worst case discriminator/generator, $\mathbf{v}_{\text{worst}} \leftarrow \arg\max_{\mathbf{v} \in \mathcal{K}_2} M(\mathbf{u}, \mathbf{v})$, and $\mathbf{u}_{\text{worst}} \leftarrow \arg\min_{\mathbf{u} \in \mathcal{K}_1} M(\mathbf{u}, \mathbf{v})$, and **(iii)** computing the duality gap by estimating: $\text{DG} := M(\mathbf{u}, \mathbf{v}_{\text{worst}}) - M(\mathbf{u}_{\text{worst}}, \mathbf{v})$. Now since we do not have direct access to the expected game objective, one should use different samples for each of the three mentioned phases in order to maintain an unbiased estimate of the expected duality gap. Thus we split our dataset into three disjoint subsets, training set, *adversary finding set*, and test set which are respectively used in phases **(i)**, **(ii)** ,and **(iii)**.

**Minimax Loss as a metric for evaluating generators.** For all experiments we report both the duality gap (DG) and the minimax loss defined as $M(\mathbf{u}, \mathbf{v}_{\text{worst}})$. The minimax loss is the first term that contributes to the computation of DG and intuitively measures the 'goodness' of a generator

---

[3]Of course, one should also use a validation set, but this is less important for our discussion here.

$G_{\mathbf{u}}$. If $G_{\mathbf{u}}$ is optimal and covers $p_{\text{data}}$, the minimax loss achieves its optimal value as well. This happens when $D_{\mathbf{v}_{\text{worst}}}$ outputs 0.5 for both the real and generated samples. Whenever the generated distribution does not cover the entire support of $p_{\text{data}}$ or compromises the sample quality, this is detected by $D_{\mathbf{v}_{\text{worst}}}$ and hence, the minimax loss increases. This makes it a compelling metric for detecting mode collapse and evaluating sample quality. Note that in order to compute this metric one only needs a batch of generated samples, i.e. the generator can be used as a black-box. Hence, this metric is not limited to generators trained as part of a GAN, but can instead be used for any generator that can be sampled from.

**Practical and efficient estimation of duality gap for GANs.** In practice, the metrics are computed by optimizing a separate generator/discriminator using a gradient based algorithm. To speed up the optimization, we initialize the networks using the parameters of the adversary at the particular step we are evaluating. Hence, if we are evaluating the GAN at step $t$, we train $\mathbf{v}_{\text{worst}}$ for $\mathbf{u}_t$ and $\mathbf{u}_{\text{worst}}$ for $\mathbf{v}_t$ by using $\mathbf{v}_t$ as a starting point for $\mathbf{v}_{\text{worst}}$ and analogously, $\mathbf{u}_t$ as a starting point for $\mathbf{u}_{\text{worst}}$ for a number of fixed steps. [4].

We also explored approximations of DG, where instead of using optimization to find the optimal $\mathbf{v}_{\text{worst}}$ and $\mathbf{u}_{\text{worst}}$, we limit the search space to a set of discriminators and generators that are stored as snapshots throughout the training, similarly to (Olsson et al., 2018) (see results in Appendix B.3).

## 5 EXPERIMENTAL RESULTS

Some notable desirable properties of a metric is that it should be able to efficiently *(i)* detect convergence and *(ii)* evaluate the quality of the generated samples. We carefully design a series of experiments to examine commonly encountered failure modes when training a GAN and analyze how this is reflected by the two metrics. Specifically, we show the sensitivity of the duality gap metric to (non-)convergence and the susceptibility of the minimax loss to reflect the sample quality.

Note that our goal here is *not* to provide a rigorous comparative analysis between different methods, but to demonstrate that both metrics capture desirable properties useful for training.

### 5.1 MIXTURE OF GAUSSIANS

We train a vanilla GAN on three toy datasets with increasing difficulty, a) RING: a mixture of 8 Gaussians, b) SPIRAL: a mixture of 20 Gaussians and c) GRID: a mixture of 25 Gaussians. As the true data distribution is known, this setting allows for tracking of convergence and mode dropping.

**Duality gap and convergence.** Our first goal is to illustrate the connection between convergence and the duality gap. To that end, we analyze the progression of the duality gap throughout training in stable and unstable settings. One common problem of GANs is the unstable mode collapse, where the generator rotates between generating different modes. We simulate such instabilities and compare them against successful GANs in Table 1. The gap goes to zero for all stable models after convergence to the true data distribution. Conversely, unstable training is reflected both in terms of the large value reached by the duality gap as well as its trend over iterations (e.g. oscillations indicate an unstable behavior). Thus the duality gap is a powerful tool for *monitoring the training and detecting unstable collapse*.

**Minimax loss reflects sample quality.** As previously argued, the duality gap achieves the lowest possible value of zero upon convergence to the NE. When $DG = 0$, the generated distribution, $p_{\mathbf{u}}$, equals the true data distribution, $p_{\text{data}}$ and there are no generated samples that lie outside of the support of $p_{\text{data}}$. Whenever $DG$ is not zero, another useful metric to look at is the minimax loss. As this measures the loss given by the most adversarial discriminator, any mode collapse or lower sample quality can be detected by $D_{\mathbf{v}_{\text{worst}}}$ and hence lead to a larger minimax loss.

For the toy datasets, we measure the sample quality using *(i)* the number of covered modes and *(ii)* the number of generated samples that fall within 3 standard deviations of the modes. The correlation of these measures with the duality gap as well as with the minimax loss is reported in Figure 1. We

---

[4]The code will be released upon acceptance.

Table 1: Progression of DG throughout training and heatmaps of the generator distribution

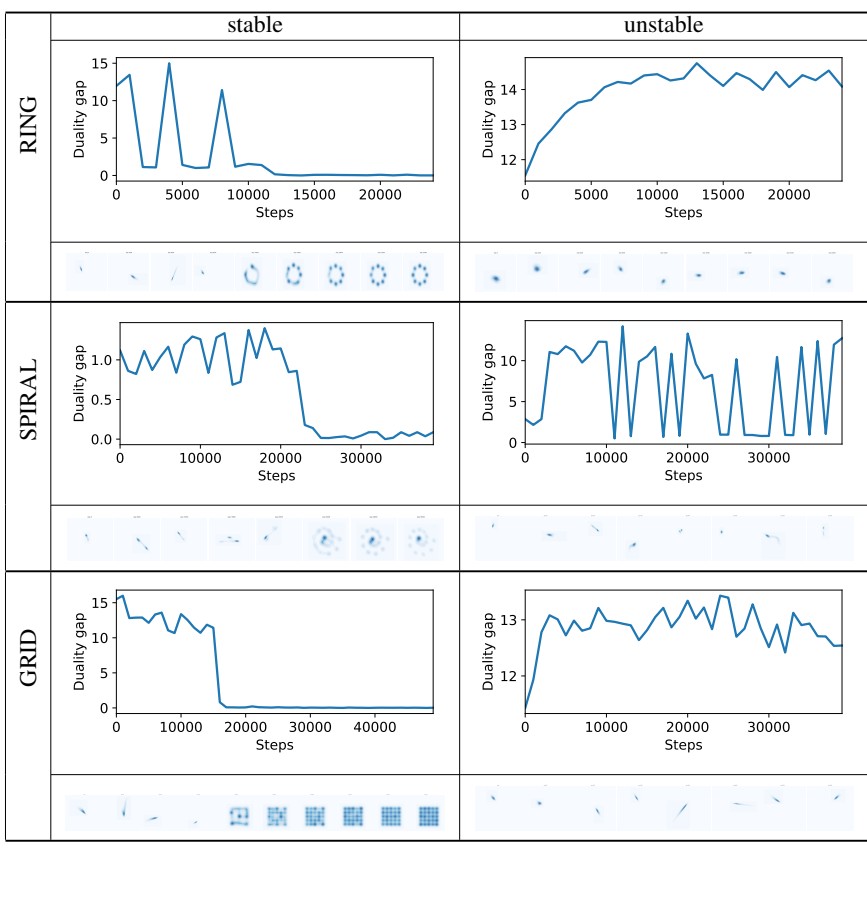

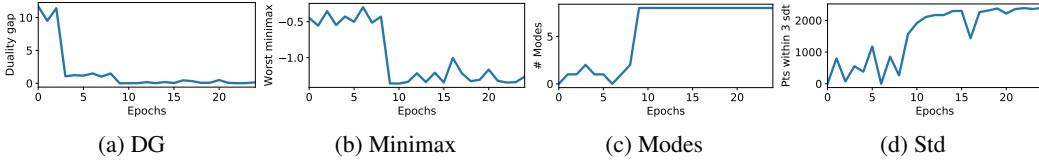

(a) DG        (b) Minimax        (c) Modes        (d) Std

Figure 1: DG, minimax, number of modes, and generated samples close to modes across epochs

observe significant anti-correlation (especially for minimax loss), which indicates that both metrics capture changes in the number of modes and generated samples that lie outside of the support of $p_{\text{data}}$, and hence *the minimax loss can be used as a proxy to determining the overall sample quality*.

## 5.2 DG AND STABLE MODE COLLAPSE

The previous experiment shows that unstable mode collapse is captured by DG. The trend of the curve is unstable and is typically within a high range. We are now interested in the case of *stable mode collapse*, where the model does converge, but only to a subset of the modes.

We train a GAN on MNIST where the generator collapses to generating from only one class (see Figure 2) (a-d) and does not change the mode as the number of training steps increases. Figure 2 e) shows the DG curve throughout the training. The trend is flat and stable, but the value of the DG is not zero, thus showing that *looking at the trend and value of the DG is helpful for detecting stable mode collapse as well*.

|  | DG | Minimax |  |
|---|---|---|---|
| Modes | -0.63 | -0.97 | ring |
|  | -0.59 | -0.93 | spiral |
|  | -0.71 | -0.95 | grid |
| Std | -0.64 | -0.94 | ring |
|  | -0.64 | -0.58 | spiral |
|  | -0.7 | -0.93 | grid |

Table 2: Pearson product-moment correlation coefficients for an average of 10 stable rounds.

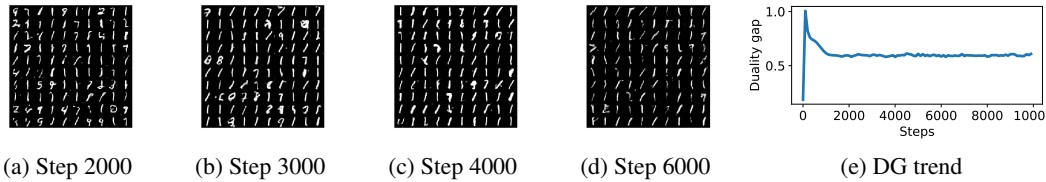

| (a) Step 2000 | (b) Step 3000 | (c) Step 4000 | (d) Step 6000 | (e) DG trend |

Figure 2: a-d: Generated samples at four different steps; e: DG through epochs

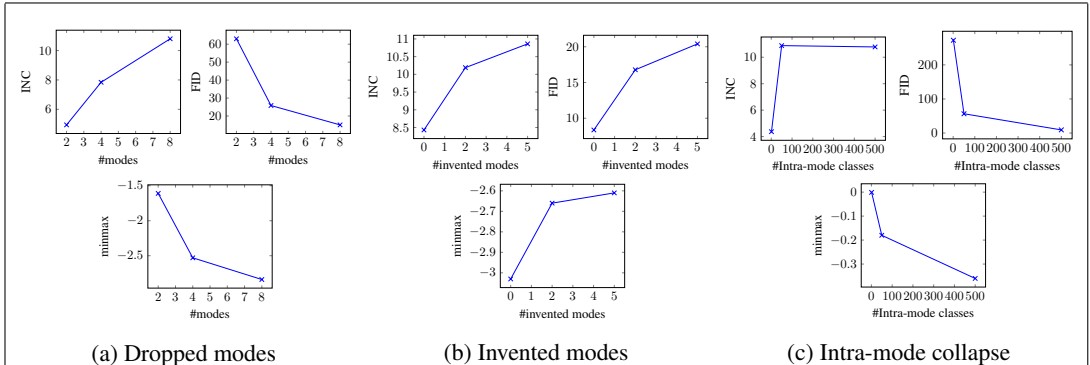

| (a) Dropped modes | (b) Invented modes | (c) Intra-mode collapse |

Figure 3: INC, FID and minimax loss for a) mode collapse (x-axis: how many modes out of 10 are generated); b) mode invention (x-axis: how many invented modes are generated) and c) intra-mode collapse (x-axis: number of unique images within a class). For INC higher is better; for FID and minimax lower is better.

## 5.3 PROPERTIES OF THE METRIC

We further analyze the sensitivity of the minimax loss to various changes in the sample quality for natural images that fall broadly in two categories: *(i)* mode sensitivity and *(ii)* visual sample quality. Using images from Cifar10 we compare against the commonly used Inception Score (INC) and Frechet Inception Distance (FID). Both metrics use the generator as a black-box through sampling. We follow the same setup for the evaluation of the minimax loss. For the computation of the minimax loss we use the standard GAN zero-sum objective. Note that changing the objective to the WGAN formulation makes the minimax loss related to the Wasserstein critic (Arjovsky et al., 2017).

**Sensitivity to modes.** The first set of experiments focuses on failures where the generated and ground truth modes do not fully overlap. As natural images are inherently multimodal, the generated distribution commonly ignores some of the true modes, which is a phenomenon known as *mode dropping*. We simulate mode dropping by using the class labels as modes. The metrics take as input a set of 5K images containing all 10 classes as 'real' images, and another set of 5K images composed of only subset of the modes (subset of 2, 4 and 8 classes) as 'generated' images (Figure 3 a).

We then turn to *mode invention* where the generator creates non-existent modes. For this setting, the set of 'real' images contains only 5 classes, whereas the sets of 'generated' images are supersets of 5, 7 and 10 classes (Figure 3 b). *Intra-mode collapse* is another common issue that occurs when the generator is generating from all modes, but there is no variety within a mode. The 'generated' sets consist of images from all 10 classes, but contain only 1, 50 and 500 unique images within a class.

Figure 3 shows the trends for all three metrics for the various degrees of mode dropping, invention and intra-class collapse. INC is unable to detect both intra-mode collapse and invented modes. On the other hand, *both FID and minimax loss exhibit desirable sensitivity to various mode changing.*

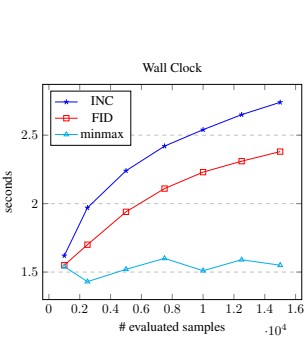

Figure 4: Wall clock time (in log-scale) for the calculation of INC, FID and minimax loss for increasing number of samples to be processed

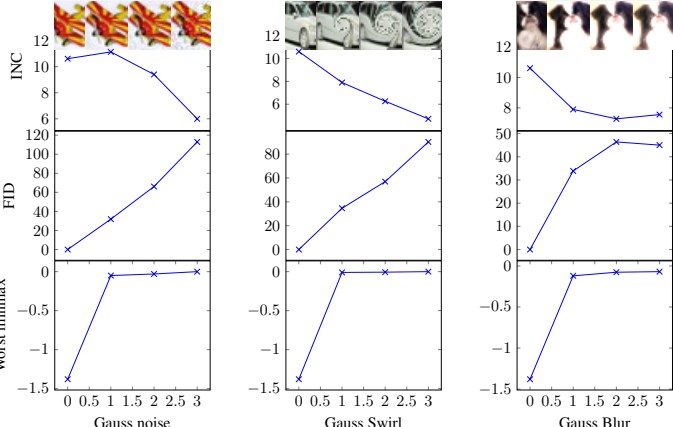

Figure 5: INC, FID and minimax loss on samples at increasing intensity of disturbance

**Sample quality.**   We now undertake an analysis of the metrics' performance on distorted samples. We distort the real images using Gaussian noise, blur and swirl at an increasing intensity. As shown in Figure 5, *all metrics, including minimax, detect different degrees of visual sample quality.*

**Efficiency.**   A metric needs to be computationally efficient in order to be used in practice to both track the progress during training and as a final metric to rank various models. Figure 4 shows the wall clock time in terms of seconds for all three metrics. We keep the number of update steps fixed to 1K which makes the computation of the minimax loss efficient, as are the other two metrics. The computation of DG takes twice the time of the computation of the minimax loss.

We also test the variance across rounds due to randomness in the seed and how this affects the final metric and overall ranking, on a simple mode collapse task. Table 3 summarizes the average of 5 rounds showing that the variance is negligible and does not affect the effectiveness of the metric.

## 5.4   GENERALIZATION TO OTHER DOMAINS AND GAN LOSSES

This experiment tests the ability of the two metrics to adapt to a different GAN loss formulation using the WGAN-GP objective (Gulrajani et al., 2017), as well as other domains. In particular, we consider the field of observational cosmology that relies on computationally expensive simulations that produce images with very different statistics from natural images. In an attempt to reduce this burden, Rodriguez et al. (2018) trained a GAN to replace the traditional N-body simulators, relying on three statistics to assess the quality of the generated samples: the mass histrogram, the peak count and the power spectral density. As explained in Rodriguez et al. (2018), these statistics are commonly used in cosmology to assess the quality of samples generated by N-body simulations. A random selection of real and generated samples shown in Figure 6 demonstrate the high visual quality achieved by the generator.

We evaluate the agreement between the statistics of the real and generated samples using the squared norm of the statistics differences (lower scores are therefore better). In Figure 7, we show the evolution of the scores corresponding to the three statistics as well as the duality gap. We observe a strong correlation, especially between the peaks. Furthermore, it seems that the duality gap takes all the statistics into account. In Figure 8, we plot the Pearson correlation between the duality gap, the minimax value and the scores. As explained in Rodriguez et al. (2018), the distribution of the raw data is long-tail and has a very high dynamic range. Hence to simplify the learning process, the data is first mapped to $[-1, 1]$ using a non-linear function. As a result, we additionally plot the same correlations for the raw data. From this experiment, we observe a strong empirical correlation between the duality gap, the minimax value and the cosmological scores.

| classes | INC | FID | Minimax |
|---------|-----|-----|---------|
| 2 classes | 4.94±0.13 | 63 | -1.69±0.06 |
| 4 classes | 7.84±0.3 | 25.85 | -2.53±0.04 |
| 6 classes | 9.88±0.18 | 18.39 | -3.14±0.03 |

Table 3: Metrics on a simple mode dropping task

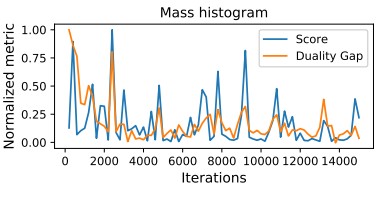
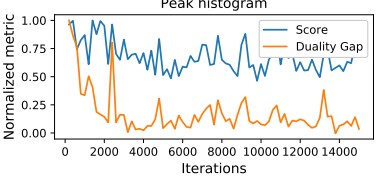
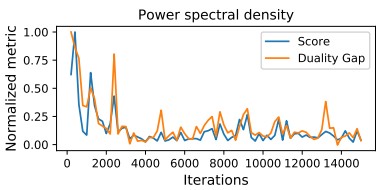

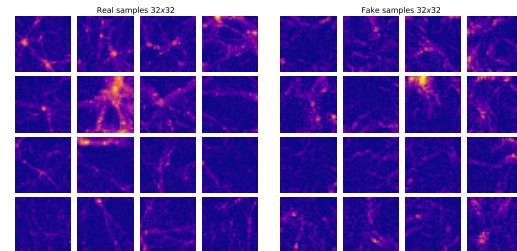

Figure 6: Real and generated cosmological images representing slices of mass density of the universe. The yellow coefficient implies a high mass concentration

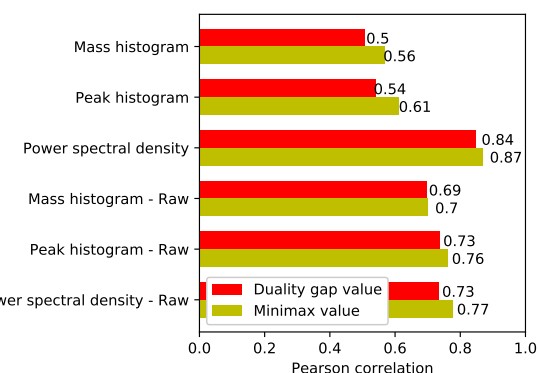

Figure 7: DG and cosmo-score evolution

Figure 8: Correlation between cosmological scores DG

## 6 CONCLUSION

The two proposed metrics complement other existing metrics and can be used by practitioners for monitoring progress towards convergence. While the minimax loss focuses on the performance of the generator (and the quality of its samples), the duality gap takes into account both the generator and discriminator. These metrics address two problems commonly faced by practitioners: 1) when should one stop training? and 2) if the training procedure has converged, have we reached the optimum or are we stuck in mode collapse? A significant advantage of the metrics is that, unlike many existing approaches, they require *no labelled data* and *no domain specific classifier*. Therefore, they are well-suited for applications of GANs other than the traditional generation task for images.

Of course, a downside is that - as most loss functions - the values obtained from these metrics are architecture and objective dependent, and can therefore not directly be compared. Yet, a practitioner can still rely on the overall trend throughout training for detecting non-convergent behaviors. Finally, one might want to use the generator as a black-box through its samples (this extends to other generative models too), in which case the minimax loss can be used as an evaluation metric directly.

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

## A    PROOF OF PROPOSITION 1

*Proof.* To simplify notation, let us denote by $p(x)$ the distribution over true samples and by $q(x)$ the distribution over fake samples generated by $G$. Let us also denote the output of the discriminator by $D(x)$. For simplicity, we will also slightly abuse notation and denote the GAN objective by $M(q, D)$. Thus, the GAN objective reads as follows,

$$M(q, D) := \frac{1}{2} \int p(x) \log D(x) dx + \frac{1}{2} \int q(x) \log(1 - D(x)) dx . \tag{5}$$

**First we prove the first part of the proposition:**    Let us first recall the definition of the Jensen-Shannon divergence of two distributions $p(\cdot), q(\cdot)$,

$$\text{JSD}(p \,||\, q) := \frac{1}{2} \text{KL} \left( p \,||\, \frac{p+q}{2} \right) + \frac{1}{2} \text{KL} \left( q \,||\, \frac{p+q}{2} \right) . \tag{6}$$

where the KL divergence is defined as,

$$\text{KL}(p \,||\, q) := \int p(x) \log \left( p(x)/q(x) \right) dx .$$

Now given a fixed solution $(q, D)$ we will show that the duality gap of this pair is bounded by the Jensen-Shannon divergence. It is well known that this divergence equals zero if both distributions are equal[5], and is otherwise strictly positive. To do so, we will first bound the minimax/maximin values for $q/D$.

**(a)** Upper Bounding Minimax Value: Given $q(x)$, the worst case discriminator is obtained by taking the derivative of the objective in Equation (5) with respect to $D(x)$ separately for every $x$ (this can be done since we assume the capacity of $D$ to be unbounded). This gives the following worst case discriminator (see similar derivation in Goodfellow et al. (2014)),

$$D_{\text{worst}}(x) := \frac{p(x)}{p(x) + q(x)} .$$

Plugging the above value into Equation (5) gives the following minimax value,

$$
\begin{aligned}
\max_D M(q, D) &= M(q, D_{\text{worst}}) \\
&= \frac{1}{2} \int p(x) \left( \frac{p(x)}{q(x) + p(x)} \right) dx + \frac{1}{2} \int q(x) \left( \frac{q(x)}{q(x) + p(x)} \right) dx \\
&= -\log 2 + \text{JSD}(p \,||\, q)
\end{aligned}
\tag{7}
$$

**(b)** Lower Bounding Maximin Value: Here we lower bound the maximin value for a given $q(x)$,

$$\min_q M(q, D) \le M(p, D) = \frac{1}{2} \int p(x) \log D(x) dx + \frac{1}{2} \int p(x) \log(1 - D(x)) dx . \tag{8}$$

Maximizing the last expression separately for every $x$ gives

$$\max_{D(x) \in [0,1]} \frac{1}{2} p(x) \log D(x) dx + \frac{1}{2} p(x) \log(1 - D(x)) dx = -\log 2$$

And plugging the above into Equation (8) gives,

$$\min_q M(q, D) \le -\log 2 . \tag{9}$$

**(c)** Upper bound on Duality Gap: Recall the definition of Duality gap,

$$\text{DualGap}(q, D) := \max_D M(q, D) - \min_q M(q, D) .$$

Using Equation (7) together with Equation (9) immediately shows that the duality gap is lower bounded by the Jensen-Shannon divergence between true and fake distributions. This concludes the first part of the proof.

---

[5]We mean equal up to sets of measure zero.

**Next we prove the second part of the proposition:**   Recall that we assume $q(x) = p(x)$. And let us take,

$$D(x) = 1/2, \qquad \forall x$$

Next we show that the Duality gap of $(G, D)$ is zero.

**(a)** Let us first compute the minimax value: Similarly to Equation (7) the following can be shown,

$$
\begin{aligned}
\max_D M(q, D) &= \frac{1}{2} \int p(x) \left( \frac{p(x)}{q(x) + p(x)} \right) dx + \frac{1}{2} \int q(x) \left( \frac{q(x)}{q(x) + p(x)} \right) dx \\
&= -\log 2 \cdot \frac{1}{2} \int (q(x) + p(x)) dx \\
&= -\log 2 .
\end{aligned}
\tag{10}
$$

where we used $p(x)/(p(x) + q(x)) = 1/2$.

**(b)** Let us now compute the maximin value. Since $D(x) = 1/2$ the following holds for any $q_0(x)$,

$$M(q_0, D) = \frac{1}{2} \int p(x) \log D(x) dx + \frac{1}{2} \int q_0(x) \log(1 - D(x)) dx = -\log 2 ,$$

which immediately implies,

$$\min_q M(q, D) = -\log 2 . \tag{11}$$

Combining Equation (10) with Equation (11), with the definition of the Duality gap implies,

$$\mathrm{DualGap(q, D)} = 0 .$$

which concludes the second part of the proof.

$\square$

# B    EXPERIMENTS

## B.1    TOY DATASET: MIXTURE OF GAUSSIANS

The toy datasets consist of a mixture of 8, 20 and 25 Gaussians for each of the models (RING, SPIRAL, GRID), respectfully. The standard deviation is set to 0.05 for all models except for the RING where the std is 0.01. Depending on the dataset, the means are spaced equally around a unit circle, a spiral or a grid.

The architecture of the generator consists of two fully connected layers (of size 128) and a linear projection to the dimensionality of the data (i.e. 2). The activation functions for the fully connected layers are relu. The discriminator is symmetric and hence, composed of two fully connected layers (of size 128) followed by a linear layer of size 1. The activation functions for the fully connected layers are relu, whereas the final layer uses sigmoid as an activation function.

Adam was used as an optimizer for both the discriminator and the generator with $beta_1 = 0.5$ and a batch size of 100. The latent dimensionality $z$ is 100. The learning rates for the reported models are given as follows in Table 4. The optimizer used for training the worst D/G is Adam and is set to the default parameters.

Plots of DG during training are given in Table 1. Table 5 lists the obtained results for the methods in terms of their final duality gap, number of modes they have covered and the number of generated points that fall within three standard deviations of one of the means. The heatmaps of the final generated distributions are given in Figure 9.

We also plot generated samples from the worst case generator in Figure 10.

Progress during training for Figure 1 is given in Figure 11.

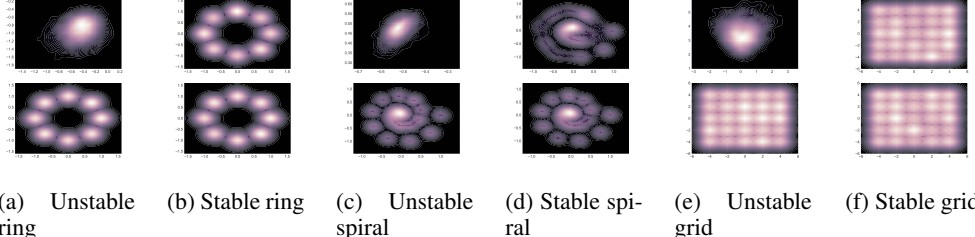

(a) Unstable ring  (b) Stable ring  (c) Unstable spiral  (d) Stable spiral  (e) Unstable grid  (f) Stable grid

Figure 9: Heatmaps of the generated distributions at the final steps. On top: trained model (stable or unstable), on bottom: $p_{\text{data}}$

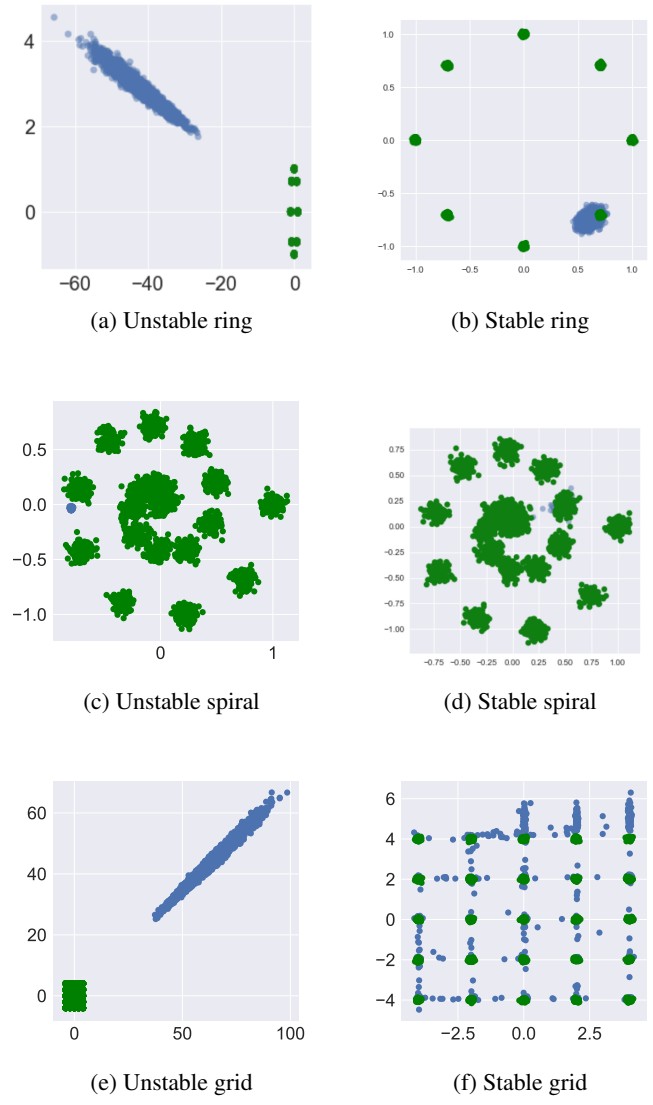

(a) Unstable ring

(b) Stable ring

(c) Unstable spiral

(d) Stable spiral

(e) Unstable grid

(f) Stable grid

Figure 10: Generated samples (in blue) from the worst generator for the discriminator for both the stable and unstable models. (ground truth in green color).

|  | stable | | unstable | |
|---|---|---|---|---|
|  | lr G | lr D | lr G | lr D |
| RING | 1e-3 | 1e-4 | 1e-4 | 2e-4 |
| SPIRAL | 1e-3 | 2e-3 | 1e-4 | 2e-3 |
| GRID | 1e-3 | 2e-3 | 1e-4 | 2e-3 |

Table 4: Learning rates used for the toy experiments.

|  |  | Duality Gap | Modes | Std |
|---|---|---|---|---|
| stable | RING | 0.04 | 8 | 2375 |
|  | SPIRAL | 0.14 | 20 | 1999 |
|  | GRID | 0.03 | 25 | 2370 |
| unstable | RING | 13 | 2 | 152 |
|  | SPIRAL | 1.22 | 1 | 1724 |
|  | GRID | 12.09 | 3 | 37 |

Table 5: Final results for DG, number of covered modes and number of generated samples (out of 2400) that fall within 3 standard deviations of the means.

### B.1.1 Loss Curves

A common problem practitioners face is when to stop training, i.e. understanding whether the model is still improving or not. See for example Figure 12, which shows the discriminator and generator losses during training of a DCGAN model on CIFAR10 (dcg). The training curves are oscillating and hence are very non-intuitive. A practitioner needs to most often rely on visual inspection or some performance metric as a proxy as a stopping criteria.

The generator and discriminator losses for our 2D ring problem are shown in Figure 13. Based on the curves it is hard to determine when the model stops improving. As this is a 2D problem one can visually observe when the model has converged through the heatmaps of the generated samples (see Table 1). However in higher-dimensional problems (like the one discussed above on CIFAR10) one cannot do the same. Figure 14 showcases the progression of the duality gap throughout the training. Contrary to the discriminator/generator losses, this curve is meaningful and clearly shows the model has converged and when one can stop training, which coincides with what is shown on the heatmaps.

### B.2 Other hyperparameters

### B.2.1 Stable Mode Collapse

The architecture of the generator consists of 5 fully connected layers of size 128 with leaky relu as an activation unit, followed by a projection layer with tanh activation. The discriminator consists of 2 dense layers of size 128 and a projection layer. The activation function used for the dense layers of the discriminator is leaky relu as well, while the final layer uses a sigmoid. The value $\alpha$ for leaky relu is set to 0.3.

The optimizer we use is Adam with default parameters for both the generator, discriminator, as well as the optimizers for training the worst generator and discriminator. The dimensionality of the latent space $z$ is set to 100 and we use a batch size of 100 as well. We train for 10K steps. The number of steps for training the worst case generator/discriminator is 400.

We use the training, validation and test split of MNIST (Deng, 2012) for training the GAN, training the worst case generator/discriminator, and estimating the duality gap (as discussed in Section 4).

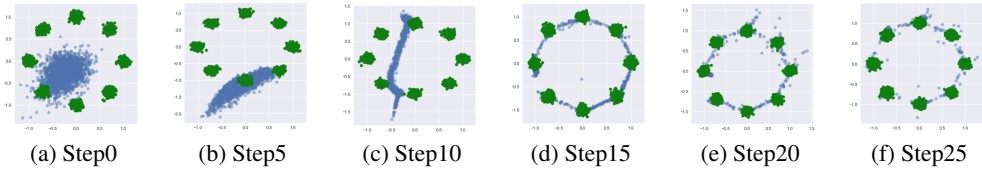

|  |  |  |  |  |  |
|---|---|---|---|---|---|
| (a) Step0 | (b) Step5 | (c) Step10 | (d) Step15 | (e) Step20 | (f) Step25 |

Figure 11: Generated samples (blue) and real samples (green) throughout training steps

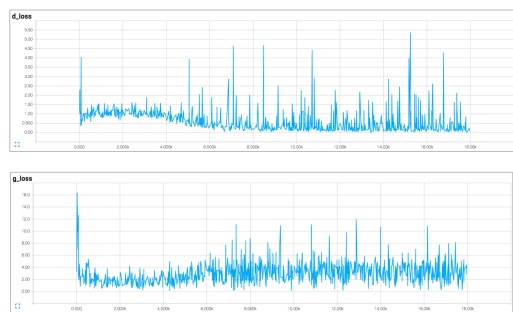

Figure 12: Discriminator and generator loss curves for a DCGAN model trained on CIFAR10. The curves are oscillating and it is hard to determine when to stop the training.

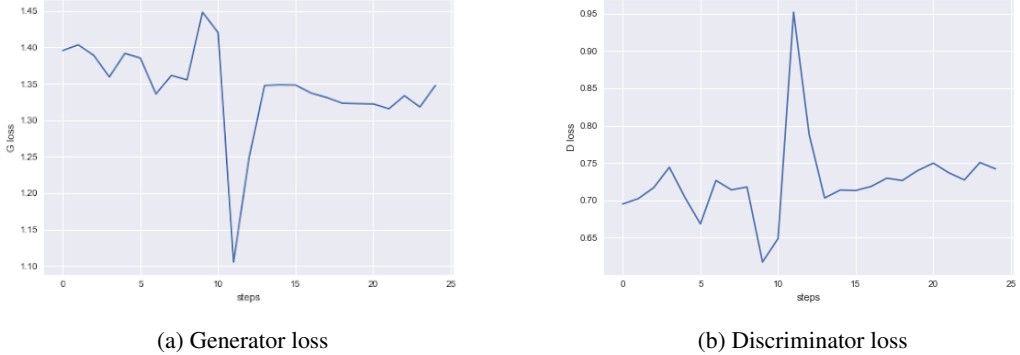

(a) Generator loss

(b) Discriminator loss

Figure 13: Discriminator and generator loss curves for the 2D ring problems. The curves are oscillating and it is hard to determine when to stop the training and when the model stops improving.

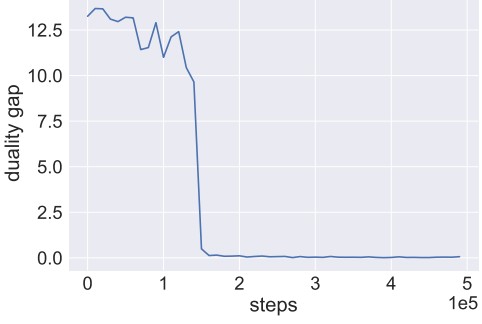

Figure 14: Curve of the progression of the duality gap during training.

### B.2.2 MINIMAX EXPERIMENT ON CIFAR10

Here we describe hyperaparameters for the experiment on Cifar10 (Krizhevsky et al., 2014). The worst case discriminator we train is using the commonly used DCGAN architecture (Radford et al., 2015). We again use Adam with the default parameters as the optimizer and the batch size is 100. We update the worst case classifier for 1K steps.

The hyperparameters used for the distortion in the experiment for visual sample quality are:

1. Gaussian noise
    (a) level 1: $sigma = 5$
    (b) level 2: $sigma = 10$
    (c) level 3: $sigma = 20$

2. Gaussian blur
    (a) level 1: $ksize = 2$
    (b) level 2: $ksize = 5$
    (c) level 3: $ksize = 7$

3. Gaussian swirl with strength 5
    (a) level 1: $radius = 1$
    (b) level 2: $radius = 2$
    (c) level 3: $radius = 20$

### B.2.3 EXPERIMENT ON COSMOLOGY DATASET

Following the approach of Gulrajani et al. (2017), we used a Wasserstein loss with a gradient penalty of 10. Both the generator and the discriminator were optimized with an "RMSprop" optimizer and a learning rate of $3 \cdot 10^{-5}$. The discriminator was optimized 5 times more often than the generator.

### B.3 APPROXIMATING THE DUALITY GAP

We explored variants in which we are circumventing the optimization in order to find the worst case generator/discriminator by using a set of discriminators/generators out of which we choose the most adversarial one. The sets are created by saving snapshots of the parameters of the two networks during training. We explored variants where the snapshots come a) only from past models and b) a mix of previous and future models, and generally found b) to perform better. This setting is similar to the models proposed in (Olsson et al., 2018), except they use skill rating systems to infer a latent variable for the successfulness of a generator. On the other hand, we compute the duality gap and the minimax loss to infer the successfulness of the entire GAN and of the generator, respectfully.

Table 6 gives the progression of the approximated duality gap for four various scenarios: stable model, unstable mode collapse and stable mode collapse. The duality gap was approximated using 10 models that spanned accross 2 epochs.

## C ANALYSIS OF THE QUALITY OF THE EMPIRICAL DG

The theoretical assumption appearing in the proof in Appendix A is that the discriminator and generator have unbounded capacity and we can obtain the true minimizer and maximizer when computing $\mathbf{u}_{\text{worst}}$ and $\mathbf{v}_{\text{worst}}$, respectively. This, however, is not tractable in practice. Furthermore, it is well known that one common problem in GANs is mode collapse. This raises the question of how the duality gap metric would be affected if the worst generator that we compute is collapsed itself. In the following we address this both empirically and from a theoretical perspective.

We use the same experimental setup as described in Appendix B.1. We focus on a GAN that has converged such that the generator covers all modes uniformly i.e. $p_g = p_{data}$ (Figure 15 a)). The discriminator outputs 0.5 for real and fake samples (Figure 15 b)). This means that the model has reached the equilibrium and the duality gap -in theory- is zero.

Table 6: Progression of DG throughout training and heatmaps of the generator distribution. A1: stable ring, A2: unstable ring, B1: mode collapse, B2: stable mode collapse

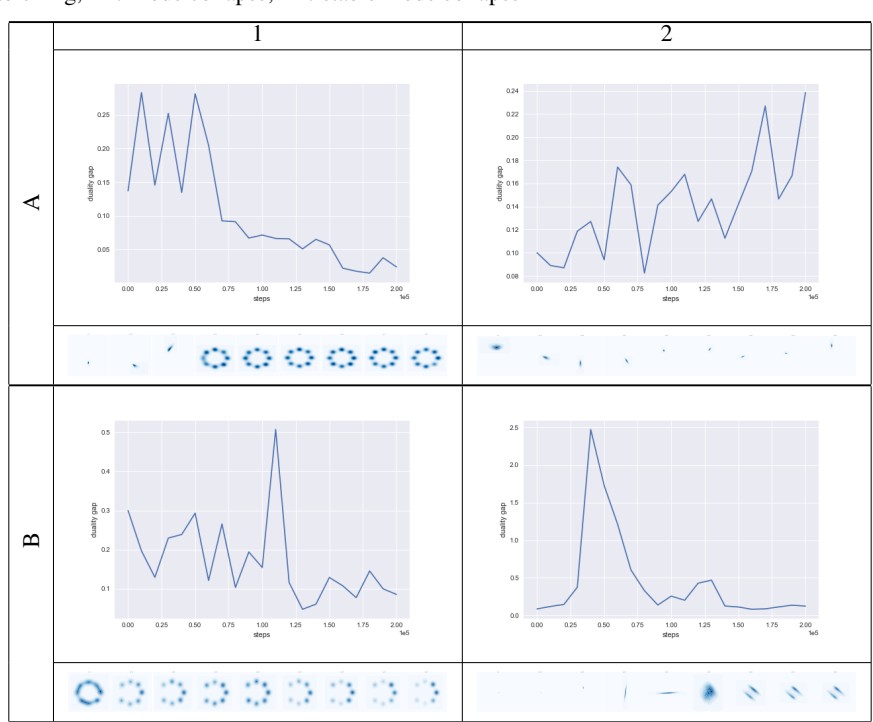

**Collapsed worst case generator.** Now we focus on the calculation of the duality gap. Let us consider the case of a mode collapsed worst case generator. In particular, when computing the maximin part of the duality gap i.e. $M(\mathbf{u}_{\text{worst}}, \mathbf{v})$, let us assume the solution was such that $G_{\mathbf{u}_{\text{worst}}}$ only covers one mode of the true distribution (Figure 15 d)). Then $M(\mathbf{u}_{\text{worst}}, \mathbf{v}) = \log(0.5) + \log(1 - 0.5)$. The minmax calculation is: $M(\mathbf{u}, \mathbf{v}_{\text{worst}}) = \log(0.5) + \log(1 - 0.5)$. Hence, the value of DG is zero, despite the collapse in the calculation for the $\mathbf{u}_{\text{worst}}$. The generator has no incentive to spread its mass due to the objective. While this is a problem for the original GAN that is being trained, it is not an issue for the calculation of the duality gap metric.

Figure 16 b) shows samples generated from $G_{\mathbf{u}_{\text{worst}}}$ when the experiment is performed in practice. We do observe that in this case, there is indeed a collapse that happened in the worst generator for the fixed GAN discriminator. Yet, $D(G_{\mathbf{u}_{\text{worst}}}) = 0.489$ and $DG = 0.002$ confirming the previous thought experiment. A heatmap with generated samples from the GAN generator are given in Figure 16.

Hence, mode collapse for the computation of DG is not an issue. Note though that when there is mode collapse in the GAN itself that is being evaluated, the DG detects this. In particular, this is supported by the high anti-correlation between DG and the number of covered modes and sample quality as shown in Table 2.

**Suboptimal solutions due to the optimization.** We now investigate the effect of the number of optimization steps used for the calculation of the duality gap on the quality of the solution. We run 5 different models with different hyperparameters with the goal to find the best setting. As suggested, we want to use the duality gap as the metric for this. Table 7 gives the results. The ranking of the models is the same for various numbers of optimization steps and corresponds to the ranking obtained by taking into consideration the number of covered modes and the number of generated samples that fall within 3 standard deviations of one of the modes.

This suggests that as long as one uses the same number of optimization steps when comparing different models, the suboptimality of the solution is empirically not an issue.

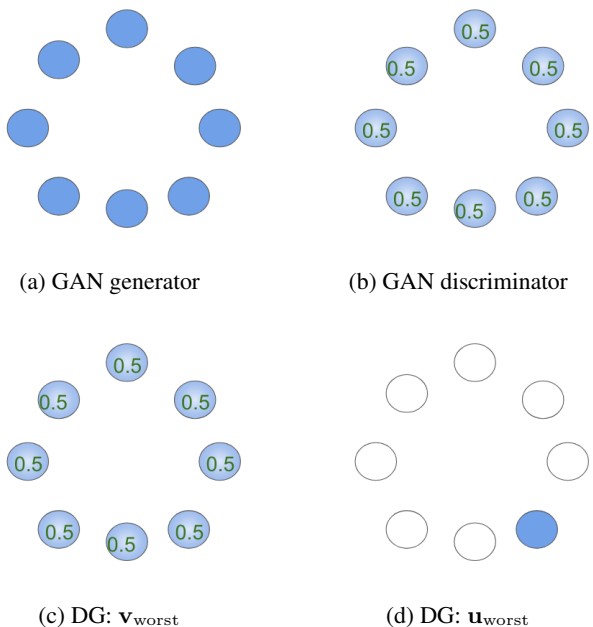

(a) GAN generator        (b) GAN discriminator

(c) DG: $\mathbf{v}_{\mathrm{worst}}$        (d) DG: $\mathbf{u}_{\mathrm{worst}}$

Figure 15: Analysis of a GAN that has reached the equilibrium for the mixture of 8 Gaussians problem. a) Samples generated from the GAN generator cover all 8 modes uniformly; b) Probabilities for a sample being real. The GAN discriminator assigns 0.5 probability to data points from the 8 modes and 0 everywhere else; c) For the computation of the duality gap, the theoretical $\mathbf{v}_{\mathrm{worst}}$ assigns 0.5 to fake/real samples for the fixed GAN generator; d) We assume there was mode collapse when computing $\mathbf{u}_{\mathrm{worst}}$ for the fixed GAN discriminator and samples from $G_{\mathbf{u}_{\mathrm{worst}}}$ lie only on a single mode.

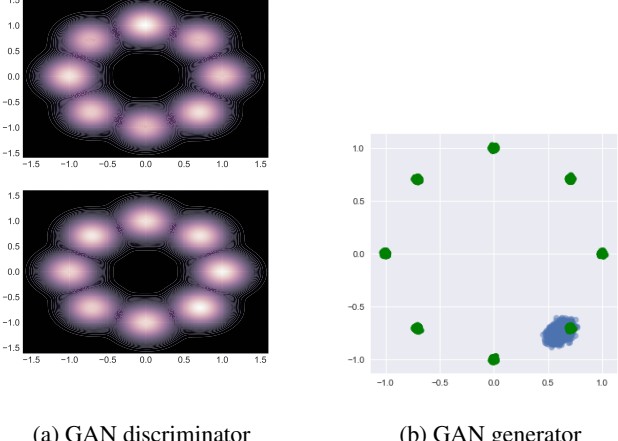

(a) GAN discriminator        (b) GAN generator

Figure 16: a) A heatmap of generated samples from the GAN generator (up) and the true data distribution (below). The generator is able to cover the true data distribution; b) Generated samples from $G_{\mathbf{u}_{\mathrm{worst}}}$.

| hyperparameters | | quality of GAN | | DG for # optimization steps | | | |
|---|---|---|---|---|---|---|---|
| lr_D | lr_G | #modes (out of 8) | # quality samples (out of 2500) | 500 | 1000 | 1500 | 2000 |
| 2e-3 | 1e-4 | 8 | 2414 | 0.014 | 0.03 | 0.04 | 0.06 |
| 1e-4 | 1e-4 | 1 | 1119 | 10.02 | 11.3 | 12.23 | 12.3 |
| 1e-3 | 1e-4 | 8 | 2440 | 0.009 | 0.01 | 0.006 | 0.02 |
| 5e-3 | 1e-4 | 8 | 2478 | 0.008 | 0.002 | 0.002 | 0.001 |
| 1e-4 | 1e-5 | 1 | 501 | 10.84 | 12.37 | 13.25 | 13.7 |

Table 7: DG for various number of optimization steps and GAN hyperparameters. The set of the best hyperparameters is the same no matter the number of optimization steps are used for the calculation of the duality gap.

