# OpenReview forum: "Evaluating GANs via Duality"
_ICLR.cc/2019/Conference_

### Official Review · AnonReviewer2 · 2018-11-02
**A nicely written work, but concerns on significance**

**Rating:** 5
**Confidence:** 3

**Review:**

This work proposes to use duality gap and minimax loss as measures for monitoring the progress of training GANs. The authors first showed a relationship between duality gap(DG) and Jensen-Shannon divergence and non-negativeness on DG. Then, a comprehensive discussion was presented on how to estimate and efficiently compute DG. A series of experiments were designed on synthetic data and real-world image data to show 1) how duality gap is sensitive to capture non-convergence during training and 2) how minimax loss efficiently reflects the sample quality from generator.


I was not very familiar with GANs, thus I'm not sure on the significance of paper and would like to see opinions from other reviews on this. For reviewing this paper, I also read the cited works such as Salimans (2016), Heusel (2017). Compared with them, the theoretical contribution of this work seems less significant. Also, I'm not quite impressed by the advantages of proposed metrics. However, this work is nicely written, the ideas are delivered clearly, experiments are nicely designed. I kind of enjoying reading this paper due to its clarity.


Other concerns:

There are two D_1 in Equation Mixed Nash equilibrium.

---

> ### Author Response · Authors · 2018-11-16
> **[Part 1/2] The metric gives a natural solution to many open challenges in GANs**
>
> Thank you for the review. We appreciate the comments on the nice flow of the paper and the carefully designed experimental section. Your two concerns are (1) “I was not very familiar with GANs, thus I'm not sure on the significance of paper” and (2) “I'm not quite impressed by the advantages of proposed metrics”, which we address below:
>
> 1. Significance
> GANs are a 2-player minimax game, which makes their objective function different than the more commonly encountered likelihood optimization problems, thus yielding new challenges in terms of optimization and evaluation. For evaluating likelihood-based models, a common metric to use is the test loss, whereas for minimax problems it is not clear what the equivalent would be [1].
> In particular, with the absence of such a metric, practitioners are facing several problems, such as (i) determining whether the model has converged, (ii) determining when to stop training (see Fig. 12 and 13), (iii) having a meaningful curve throughout training as the discriminator and generator losses are not intuitive, (iv) comparing different runs and (v) debugging the model in the sense of (un)stable mode collapse, non-convergence etc. See for example [1]: “Generative adversarial networks are not born with a good objection function that can inform us about the training progress. Without a good evaluation metric, it is like working in the dark. No good sign to tell when to stop; No good indicator to compare the performance of multiple models.”. Current methods for stopping criteria rely on visual inspection and/or using some sample-quality metric as a proxy. However this is not principled and it is unclear how to compute the metrics in non-image domains. Thus another open challenge with GANs is (vi) having a useful metric that is domain independent.
>
> In this work, we argue that such a natural metric exists, namely the duality gap and its minimax part. We show how to compute it in practice in an efficient way and demonstrate its desirable properties across different GAN pitfalls, domains and GAN objectives. Thus, in our opinion, this work is of significance for the GAN community, not only for practical purposes as it gives a solution to the previously mentioned open problems (i-vi) both in theory and practice, but also from research perspective as it gives a reliable non-convergence metric to help analyse which methods actually converge, which is one of the central issues of GANs. Note that current practical analyses mainly focus on 2-dimensional problems where the solution can be visually inspected due to the lack of such a metric [2].
>
> 2. Advantages of the proposed metrics
> From a theoretical perspective, the DG is very natural for the detection of non-convergent behaviors, it is always non-negative and is zero if and only if the model has reached a (Nash) equilibrium.
>
> The experimental results presented in the paper provide a thorough evaluation of the metric introduced in our submission. We included various tests that focus on common pitfalls encountered with GANs and demonstrated that the proposed metric can detect these corner cases. In particular:
>
> In experiment 5.1 we demonstrate that the *DG yields a meaningful curve* throughout training and detects convergent and non-convergent behaviours. Please note that the commonly used metrics such as FID and Inception score cannot be applied to these datasets.
>
> In experiment 5.2 we show that the *DG detects stable mode collapse* and can distinguish between stable and unstable collapses.
>
> In experiment 5.3 we empirically demonstrate that the *minimax metric detects visual sample quality (adding noise, Gaussian swirl and blur) and is very sensitive to change of modes* (mode dropping, mode invention and intra-mode collapse). It works better than Inception score, and as well as FID. However, both the Inception score and FID rely on a pre-trained Imagenet classifier, whereas our metrics need no labeled data or a pre-trained classifier.
>
> Finally, in experiment 5.4 we show the *DG metric can be applied on another GAN minimax formulation (WGAN) and on another domain that is not natural images (cosmology data)*. We find that the metric is highly correlated with a domain specific measure of performance used in cosmology. Note that the domain-specific metric requires expert knowledge and its computation is very slow, unlike the DG. Furthermore, the Inception score and FID cannot be applied on this data as they require an imagenet classifier (i.e. trained with labeled natural images).

---

> > ### Author Response · Authors · 2018-11-16
> > **[Part 2/2] The metric gives a natural solution to many open challenges in GANs**
> >
> > We summarize the properties and advantages of our approach in the table shown below, including a comparison to Inception score (INC) and FID.
> >
> > +---------------------------------------------------------------------------------------------+----------------+--------+--------------+
> > | Property\Metric                                                                                      | INC            | FID   | minimax |
> > +---------------------------------------------------------------------------------------------+----------------+--------+--------------+
> > | Sensitivity to mode collapse                                                                | moderate | high | high         |
> > +---------------------------------------------------------------------------------------------+----------------+--------+--------------+
> > | Sensitivity to mode invention                                                              | low            | high | high         |
> > +---------------------------------------------------------------------------------------------+----------------+--------+--------------+
> > | Sensitivity to intra-mode collapse                                                       | low            | high | high        |
> > +---------------------------------------------------------------------------------------------+----------------+--------+--------------+
> > | Sensitivity to visual quality and transformations                             | moderate | high | high        |
> > +---------------------------------------------------------------------------------------------+----------------+--------+--------------+
> > | Computational: Fast                                                                              | yes             | yes   | yes          |
> > +---------------------------------------------------------------------------------------------+----------------+--------+--------------+
> > | Computational: Needs labeled data or a pretrained classifier      | yes             | yes   | no           |
> > +---------------------------------------------------------------------------------------------+----------------+--------+--------------+
> > | Computational: Can be applied to any domain without change  | no               | no    | yes          |
> > +---------------------------------------------------------------------------------------------+----------------+--------+--------------+
> >
> >
> > We hope to have addressed your concerns and that our reply is detailed and informative enough so that the reviewer can reconsider their judgement. We are looking forward to the reply.
> >
> > References:
> > [1] Ermon et al. Generative Adversarial Networks, [cs236, Stanford], <http://cs236.stanford.edu/assets/slides/cs236_lecture9.pdf#page=19>
> > [2] Lil’Log https://lilianweng.github.io/lil-log/2017/08/20/from-GAN-to-WGAN.html#lack-of-a-proper-evaluation-metric
> > [3] Mescheder et al. Which Training Methods for GANs fo actually converge? [ICML 2018] arXiv:1801.04406

---

### Official Review · AnonReviewer3 · 2018-11-02
**usage of duality in GANs, moderate contribution**

**Rating:** 3
**Confidence:** 4

**Review:**

The focus of the submission is GANs (generative adversarial network), a recent and popular min-max generative modelling approach. Training GANs is considered to be a challenging problem due to the min-max nature of the task. The authors propose two duality-inspired stopping criteria to monitor the efficiency and convergence of GAN learning.

Though training GAN can have some useful applications, the contribution of the submission is pretty moderate.
i) Duality-inspired approaches, embedded also in optimization have already been proposed: see for example 'Xu Chen, Jiang Wang, Hao Ge. Training Generative Adversarial Networks via Primal-Dual Subgradient Methods: A Lagrangian Perspective on GAN. ICLR-2018.'.
ii) The notion of generator and discriminator networks with unbounded capacity (which is an assumption in 'Proposition 1') lacks formal definition. I looked up the cited Goodfellow et al. (2014) work; it similarly does not define the concept. Based on the informal definition it is not clear whether they exist or are computationally tractable.

Minor comments:
-MMD is a specific instance of integral probability metrics when in the latter the function space is chosen to be the unit ball of a reproducing kernel Hilbert space; they are not synonyms.
-mixed Nash equilibrium: E_{v\sim D_1} should be E_{v\sim D_2}.
-It might be better to call Table 1 as Figure 1.
-References: abbreviations and names should be capitalized (e.g., gan, mnist, wasserstein, nash, cifar). Lucic et al. (2017) has been accepted to NIPS-2018.

---

> ### Author Response · Authors · 2018-11-16
> **Our usage of duality in GANs is in terms of evaluation, not optimization**
>
> Thank you for reviewing our paper. Please find our replies inline:
>
> 1. “Though training GAN can have some useful applications, the contribution of the submission is pretty moderate.”
> First, we would like to stress that the contribution of our paper is not to train GANs. Instead, our contribution is to propose a reliable convergence metric for GANs that can be computed efficiently. The need for such a convergence metric has been pointed out for example in the context of analysing convergence of GANs [1] and understanding when to stop the GAN training [2, 3] (see also Fig. 12 and 13). Indeed, most empirical convergence analyses are for 2-dimensional problems due to the lack of such metric (See for example Mescheder et al. [ICML 2018]: “Measuring convergence for GANs is hard for high dimensional problems, because we lack a metric that can reliably detect non-convergent behavior. We therefore first examine the behavior [...] on simple 2D examples where we can assess convergence using an estimate of the Wasserstein-1-distance.” [1]). Furthermore, since GANs are framed as a 2-player minimax game the stopping criteria is unclear in comparison to the more traditional likelihood training [2]. In this work we argue that there is a convergence metric suitable for the general GAN game.
> In particular, our main contributions are:
> - Proposing the duality gap as a natural convergence metric and the minimax metric as a performance metric in GANs
> - Show how an unbiased estimate of the metrics can be efficiently computed in practice without slowing down training
> - Design experiments that target all of the common pitfalls of GANs (stable and unstable mode dropping/invention, intra-mode collapse, non-image domain, distortion of visual quality etc.) and demonstrate empirically that the metrics are able to capture and detect all of those
>
> Thus the two metrics show very desirable properties both in theory and practice. We believe that the DG metric is very helpful as a monitoring tool for any practitioner training a GAN. The benefits are: (i) knowing whether the model has converged; (ii) knowing when to stop training; (iii) have a meaningful curve throughout training that reflects the performance of the model (i.e. whether it’s improving or not); (iv) comparison of different runs and hyperparameter searches and (v) debugging. As computing the metric is very efficient in practice this comes at no significant computational cost, and unlike other metrics requires no labels or a pre-trained classifier and can be applied to any minimax GAN formulation and any domain as demonstrated empirically.
> Further, it allows for pushing the research of the non-convergence issue on GANs on problems that are beyond 2-dimensional where they can visually be analysed. We have updated the write-up to make our contribution clearer, both with respect to existing work, as well as the importance of a convergence metric for the community.
>
> 2. “Duality-inspired approaches, embedded also in optimization have already been proposed”
> Although the reference cited by the reviewer does discuss duality for GANs, it does so in a very different context since it discusses a Lagrangian view to train GANs while we are interested in using the duality GAP as a convergence measure (and not as a training criterion). One problem we focus on in our submission is to demonstrate how to efficiently compute such measure during training, we therefore do not modify the training objective. We added a brief discussion in the related work section.
>
> 3. “The notion of generator and discriminator networks with unbounded capacity (which is an assumption in 'Proposition 1') lacks formal definition”
> As noted by the reviewer, we re-used the notion originally introduced in the GAN paper. Informally, we consider the capacity as the flexibility of a model to learn a variety of functions. More formally, we regard the capacity as the size of the space that can be approximated with the generator and discriminator. In most cases, neural networks are universal approximators and can therefore approximate any function (i.e. they are dense in the target space), thus leading us to assume they have “unbounded capacity”.
>
> We hope that we have cleared out any confusion and are looking forward to the reviewer’s reply.
>
> References:
> [1] Mescheder et al. Which Training Methods for GANs fo actually converge? [ICML 2018] arXiv:1801.04406
> [2] Chiu et al. GAN Foundations, [CSC254, University of Toronto], <https://www.cs.toronto.edu/~duvenaud/courses/csc2541/slides/gan-foundations.pdf#page=9>
> [3] Ermon et al. Generative Adversarial Networks, [cs236, Stanford], <http://cs236.stanford.edu/assets/slides/cs236_lecture9.pdf#page=19>

---

### Official Review · AnonReviewer1 · 2018-11-04
**Please justify the novelty and validation, and explain the computation details**

**Rating:** 4
**Confidence:** 3

**Review:**

In this paper, the authors proposed the duality gap as the criterion for evaluating the training of GAN. To justify the proposed criterion, the authors designed empirical experiments on both synthetic and real-world datasets to demonstrate the ability of the duality gap for detecting divergence, mode collapse, sample equality, as well as the generalization to other application domains besides image generation. Comparing with the existing criteria, e.g., FID and INC, the duality gap shows better ability and computational efficiency.


However, the paper ignores rich literatures in optimization that uses the duality gap as the criterion for characterizing the convergence of algorithms for min-max saddle point problem, e.g., [1]. In fact, in optimization community, using duality gap to screening the convergence on saddle point problem is a common knowledge. [1] even provides the finite-step convergence rate when the saddle point problem is convex-concave. This paper is only introducing that into machine learning community. Therefore, the novelty of the paper seems not enough.

Secondly, the duality gap is only able to screen the optimization convergence and the solution quality w.r.t. **the same objective**. It is not valid to compare different GANs with different losses function using the duality gap. Theoretically, for any loss function derived from some divergences, e.g. [2], the global optimal solution  can always achieve zero duality gap. In other words, for different GANs, with different objectives, the duality gap cannot distinguish which one is better. In such sense, the title is very misleading.

Thirdly, how the evaluate such criterion in practice in GAN scenario is not clearly explained. Considering the neural network parametrization of both the generator and discriminator, the argmax_v M(u, v) and argmin_u M(u, v) is not tractable. Without the optimal solution, what is the meaning of the ``duality gap'' should be explained. What will happen if we only obtain the suboptimal solutions which themselves are model collapsed? Without such discussion in both theoretical and/or empirical aspects, I am not very convincing about the conclusion.

Finally, if one follows the Fenchel dual view of GAN in [2, 3], the min-max is the variational form of some divergences, which the GANs are directly optimizing. It is straightforwardly to see the better min-max value is, the smaller divergence between generated samples and ground-truth is, and thus, the better quality of the generator is. The fact that min-max objective is indeed able to characterize the quality of generator is obvious and well-known. Otherwise, there is not need to use such objective in the optimization to train the model.


[1] Nemirovski, A., Juditsky, A., Lan, G., and Shapiro, A. (2009). Robust stochastic approximation approach to stochastic programming. SIAM J. on Optimization, 19(4):1574–1609.

[2]  I. Goodfellow, J. Pouget-Abadie, M. Mirza, B. Xu, D. Warde-Farley, S. Ozair, A. Courville, and Y. Bengio. Generative adversarial nets. In Advances in Neural Information Processing Systems (NIPS), pages 2672–2680, 2014.

[3] S. Nowozin, B. Cseke, and R. Tomioka. f-gan: Training generative neural samplers using variational divergence minimization. arXiv:1606.00709, 2016

---

> ### Author Response · Authors · 2018-11-16
> **[Part 1/2] An important open issue in GANs is the lack of a convergence metric: We introduce the duality gap to GANs and show how to compute and use it efficiently in practice**
>
> We thank the reviewer for the thoughtful comments. In the following we address their concerns and questions:
>
> 1. “Please justify the novelty and validity”
> First, we would like to emphasize that the lack of a convergence metric for GANs is an open issue in the community. As discussed in the introduction, the need for such a metric is crucial and affects several important aspects such as:
>
> - *Convergence analysis*. Over the past years, GANs have been the subject of intense research in the community, giving rise to a plethora of GAN models as well as training methods. In practice, it has been observed that GANs might not converge under certain settings. While this non-convergence behavior can in practice be visually recognized for some low-dimensional examples (such as a 2D mixture of Gaussian), this is in general more difficult in high-dimensional spaces due to the lack of a convergence metric. This problem is actually often discussed in the litterature, see e.g. Mescheder et al. [ICML 2018]: “Measuring convergence for GANs is hard for high dimensional problems, because we lack a metric that can reliably detect non-convergent behavior. We therefore first examine the behavior [...] on simple 2D examples where we can assess convergence using an estimate of the Wasserstein-1-distance.” [1]
> - *Stopping criteria and meaningful curve*. GANs are known to be hard to train in practice [2]. One of the common challenges practitioners are facing is when to stop training. See for example [3]: “GAN foundations: cons: Unclear stopping criteria”. In particular, it is well known that the curves of the discriminator and generator losses oscillate and are non-informative as to whether the model is improving or not (See Fig. 12 and 13). This is especially troublesome when a GAN is trained on non-image data in which case one might not be able to use visual inspection or FID/Inception score as a proxy.
> - *Domain-independent evaluation metric*. Commonly used evaluation metrics such as FID and Inception Score are mainly suitable for natural images as they rely on a pretrained Imagenet classifier. This is also a problem that is commonly discussed in the literature, see e.g. [4]: “Generative adversarial networks are a promising [...] that has so far been held back by unstable training and by the lack of a proper evaluation metric.”. Instead the metric suggested in the paper is does not require any specific type of data and was for example shown empirically to generalize to cosmological data.
>
> Hence, the metric we propose is a more generic tool that can serve as a) a monitoring tool to help practitioners throughout training, b) a domain-independent metric that can help spread the use of GANs to non-image domains.
>
> The duality gap (DG) and the minimax value are natural metrics for this, as they are well known to capture exactly that. As rightfully pointed out by the reviewer, the duality gap is a well-known notion in optimization and our contribution is its introduction as a metric for GANs. An important aspect we discuss in the paper is with regard to an efficient way to estimate the duality gap without slowing down training. Note that although the two metrics may seem “too natural” from an optimization point of view, they are simply **not** used in the community, despite the need for them as we discussed earlier.
> See for example Salimans et al: “Generative adversarial networks lack an objective function, which makes it difficult to compare performance of different models.” [4] and “GAN optimization challenges: No robust stopping criteria in practice (unlike likelihood based learning)” [5]. In this work, we argue that such a metric does exist and it indeed comes naturally from the objective function. This is also what our experiments demonstrate.
>
> 2. “The paper ignores rich literatures in optimization...”
> Yes, we do agree, but note that (to the best of our knowledge) almost all the existing literature focuses on solving minimax problems with convex-concave objectives and therefore existing proof guarantees do not apply to GANs. Our contribution does not relate to optimising GANs, but instead in showing that the duality gap can be empirically computed and yields good estimates of the convergence of a GAN. We revised the text to clearly emphasize this and also included the suggested reference.

---

> > ### Author Response · Authors · 2018-11-16
> > **[Part 2/2] An important open issue in GANs is the lack of a convergence metric: We introduce the duality gap to GANs and show how to compute and use it efficiently in practice**
> >
> > 3.  “...DG is only able to screen the optimization convergence and the solution quality w.r.t. the same objective…”
> > Yes, we did discuss this aspect in the conclusion. Note that convergence curves could potentially be normalized but this requires further investigation that we plan on doing as a future work. Given a fixed objective, the DG yields a curve that can be used for: debugging, hyperparameter tuning, understanding whether the model has converged or whether it suffers from stable or unstable collapse, and of course, as mentioned earlier, it serves as a stopping criterion.
> >
> > Further, a recent interest in the field is to understand which regularizer stabilizes GAN training by keeping the objective fixed and changing the regularizer [1, 6, 7]. This is yet another example where the DG would be meaningful for exactly examining the effect of the regularizer on top of a meaningful curve. Hence, the metric, as is, is useful both in practice and for pushing the research efforts forward.
> >
> > 4. “how to evaluate such criterion in practice in GAN?”
> > Please refer to Section 4 “Estimating the DG metric for GANs”, where we explain the details of the practical computation. We also include some subtleties on how to accurately use train/val/test set in order to get an unbiased estimate for the estimation of the DG for GANs.
> > “Without the optimal solution, what is the meaning of the ``duality gap'' should be explained. (theory/empirical) What will happen if we only obtain the suboptimal solutions which themselves are model collapsed?”.
> > Thank you for raising this point. We added a section to the appendix addressing this question. Both theoretically and empirically we analyse how the suboptimality of the solution affects its quality. In particular, we focus on (a) the case where the worst generator used for the computation of the maxmin of DG is itself collapsed, and (b) investigate how the number of optimization steps affects the solution. In summary, we find that this is not an issue, both in terms of theory and practice. Please see Appendix Section C for more details.
> >
> > 5. “the min-max is the variational form of some divergences, which the GANs are directly optimizing”
> > The estimation of divergences is difficult, whereas we show we can efficiently approximate DG.
> >
> > To conclude, based on the review we have updated the manuscript to more clearly emphasize its contribution which we believe was the main concern raised by the reviewer.
> >
> > References:
> > [1] Mescheder et al. Which Training Methods for GANs fo actually converge? [ICML 2018] arXiv:1801.04406
> > [2] Soumith Chintala. How to train a GAN?, NIPS Tutorial, 2016
> > [3] Chiu et al. GAN Foundations, [CSC254, University of Toronto], <https://www.cs.toronto.edu/~duvenaud/courses/csc2541/slides/gan-foundations.pdf#page=9>
> > [4] Salimans et al. Improved Techniques for Training GANs [NIPS 2016] arXiv:1606.03498
> > [5] Ermon et al. Generative Adversarial Networks, [cs236, Stanford], <http://cs236.stanford.edu/assets/slides/cs236_lecture9.pdf#page=19>
> > [6] Fedus et al. Many Paths to Equilibrium: GANs Do Not Need to Decrease a Divergence At Every Step. [ICLR 2018], arXiv:1710.08446
> > [7] Kurach and Lucic et al. The GAN Landscape: Losses, Architectures, Regularization, and Normalization. arXiv:1807.04720

---

> > > ### Comment · AnonReviewer1 · 2018-11-29
> > > **Two issues are still needed to be addressed**
> > >
> > > I understand the importance of convergence analysis and stopping criterion for GAN and I acknowledge the duality gap has not been introduced in the GAN community. However, this is *NOT* the reason to ignore the existing literature on purpose. I am glad the authors add the related references.
> > >
> > > I think the following two issues are still needed to be addressed:
> > >
> > > 1, The term "evaluation" seems mis-interpreted by the authors. As the authors agreed in the reply, the dual gap is only valid for the fixed objective. It is meaningless to use such criterion for "evaluating" different GANs with different objectives, i.e., used for comparing difference GANs. From this sense, the duality gap is different from FID and inception score. It will be better if the authors switch to other terminology rather than saying "evaluation GANs".
> > >
> > > 2, My question "how the evaluate such criterion in practice in GAN scenario is not clearly explained" is not answered satisfiedly. I carefully read the Appendix C about the optimality issue in computing the true duality gap. I cannot agree with the claim "This suggests that as long as one uses the same number of optimization steps when comparing different models, the suboptimality of the solution is empirically not an issue". Leave the empirical results are only on synthetic dataset, such claim is even conflict with the motivation of the paper. Actually, with an even weaker assumption that one can obtain the epsilon-optimal solution to the discriminator, [1] already shows the SGD algorithm converges to the stationary point. There is no need to compute the duality gap for screening the convergence. This is in fact, an extremely strong assumption!
> > >
> > > Consider the difficulty of obtaining the global optimum, I think it will be good enough if the author can characterize the meaning of the ``duality gap'' without the optimal solution. Otherwise, only introducing the "duality gap" concept from convex-concave optimization is not significant for a separate paper.
> > >
> > >
> > > [1] Maziar Sanjabi, Jimmy Ba, Meisam Razaviyayn, and Jason D. Lee. On the Convergence and Robustness of Training GANs with Regularized Optimal Transport. NIPS 2018.

---

> > > > ### Author Response · Authors · 2018-12-04
> > > > **Practical evaluation of the duality gap**
> > > >
> > > > 1) Thank you for your suggestion. We will change the term "evaluation" into "monitoring" .
> > > >
> > > > 2) There are two good reasons to use the duality gap as we compute it in practice:
> > > >     a) If we compute an approximate worst case D/G (say up to some \epsilon), then this affects the duality-gap up to a factor of \epsilon. Therefore, an approximate solution translates to an approximate duality gap.
> > > >     b) As we show in the experiments, our approximation to the duality gap actually gives good handle on the "quality" of the generator.
> > > >         This is not limited to toy examples, but rather also to real-world settings, such as MNIST and cosmological images (as we describe in section 5).

---

### Public Comment · ~Frans_A_Oliehoek1 · 2018-11-21
**duality = exploitability?**

I actually think that this paper is on the right track to propose a measure of convergence to GANs. So much in fact, that we have proposed the same measure, which we call exploitability, published at BNAIC/BeNeLearn:
https://bnaic2018.nl/wp-content/uploads/2018/11/Benelearn_2018_paper_3.pdf

An extensive discussion of 'exploitability' us given in an extended arXiv version:
https://arxiv.org/abs/1806.07268

I would happily discuss any potential differences (e.g., we formulate GANs in terms of mixed strategies inherently), but my impression is that the notions are the same?

---

> ### Author Response · Authors · 2018-11-22
> **We discuss the differences below**
>
> Thank you for pointing out your relevant paper, which we will cite in the revised version. The exploitability measure is indeed the mixed strategy formulation of the duality gap metric. We do agree that the metric is a very natural metric for convergence in minimax games, and is well known in optimization as also pointed out by Reviewer 1.
>
> Some of the notable differences are:
>
> 1. *Stochasticity*. It seems that your work does not take into account the stochasticity of GANs. This aspect makes the computation of our metric more difficult as there are subtleties on how one needs to use the following 3 (disjoint) sets: a) training, b) adversary finding and c) test set in order to obtain an unbiased estimate. We discuss this in detail in Section 4.
>
> 2. *Practical computation*. One crucial aspect of our work is to discuss an efficient practical computation of the metric for GANs. We suggest to initialize the models with the last version of the generator/discriminator, which makes the optimization more efficient. We also empirically demonstrate its efficiency in terms of computation time. We also explore a further approximation by using snapshots from the history.
>
> 3. *Empirically demonstrating the desirable properties of the metric/Showing the metric works*. While in your paper, you evaluate the algorithms using the exploitability metric, we evaluate the evaluation method i.e. the metric. We did extensive experiments showcasing the duality gap metric detects convergent and non-convergent behavior, stable mode collapse, sample quality and can be applied to any domain and any minimax GAN formulation (e.g. WGAN).
>
> 4. *Demonstrating how a practitioner can use the metric*. We demonstrate how the curves look like in specific GAN scenarios, and show how the metric can be used as a monitoring, debugging and tuning tool.
>
> 5. *Large scale experiments and comparison to baselines*. In our work we perform an extensive experimental study on the following real datasets: CIFAR10, MNIST and a cosmology dataset. Furthermore, we compare against commonly used strong baselines (FID and Inception score) and discuss the differences. We also compare against domain-specific metrics developed by experts for the cosmology dataset and show high correlation.
>
> 6. *Discussing cons and suboptimality of the approximation*. As the practical solution is an approximation of the theoretical metric, we discuss how the suboptimality of the solution affects the quality of the score. In particular, we evaluate and discuss what happens when the most adversarial G for a fixed D collapses. We believe these are important aspects for the properties of the practical version of the metric. At the same time, we also show that duality-gap gives a direct handle to mode collapse. This is formalized in Proposition 1 of our paper.
>
> Again, thank you for highlighting your paper. We will add a discussion emphasizing the differences between the two approaches in the revised version.

---

> > ### Public Comment · ~Frans_A_Oliehoek1 · 2018-11-24
> > **would like to see this published, but some formulations could be adapted**
> >
> > Thank you for discussing.
> >
> >
> > I think 3,4,5 are very fair points. Indeed I am quite excited about seeing a thorough evaluation of this metric!
> >
> > The discussion of different data partitions (point 1), is useful. Since we were using with a synthetic mixture of Gaussian tasks only, this was not an issue for us: we could simple generate new point for all these on-the-fly.
> >
> > However, I am unsure what you mean by "it seems that your work does not take into account the stochasticity of GANs". I cannot imagine how one could have a GAN that is not stochastic? So we certainly deal with that stochasticity. (Also, we tackle the GAN using mixed strategies, if that is what is intended)?
> >
> > One of the points I object to is (6). I cannot find the location in the paper where the sub-optimality of of the practically applied algorithm is discussed? Actually, I think that most of the formulations in your submission seem to suggest that you can compute the duality gap (and even efficiently!). Of course, this is not the case: computing the duality GAP requires computing 2 best responses (solving 2 non-convex optimization problems), and we can not find the optimal solutions in general. In contrast, our paper is extremely up front about this: this is why we introduce "resource bounded best responses", which provide as good as a best response as one can compute given finite resources.
> >
> > As such, I believe what you actually compute is what one could call the "resource bounded duality GAP (RB-DG)", which is precisely our measure of (resource bounded) exploitability, eq. (11) in our arxiv paper https://arxiv.org/abs/1806.07268 ?
> >
> > As for point 2, I think proposing such techniques is useful, but it is not quite clear to me where the merit of these techniques are evaluated. As above, I think it could be useful to reformulate the terminology, though. Really "practical and efficient estimation of duality gap for GANs" does (as far as we know) not exist?
> >
> >
> > I think that in terms of motivating the metric, there are some point that we cover in section 6 / appendix B.2 of our arxiv paper, that could be useful to adopt:
> >
> > -the reason why worst case generator performance is not directly useful, is because we do not know that value of the game (only in the infinite capacity setting the value of v*=log 4 hold, for a finite parametric setting this value is unknown, however). As such, these metrics would allow comparing different generators, but are not useful for knowing if one is far from an equilibrium. (even if one could compute this quantity exactly!)
> >
> > -I think exploitability is extremely important to be certain about the performance of the generator:
> > "In particular, the exploitability of the classifier actually may provide information about the quality of the generator: if the generator holds up well against a perfect classifier, it should be close to the data distribution."
> >
> > -I actually disagree with a statement in the conclusion of your paper
> > "Of course, a downside is that - as most loss functions - the values obtained from these metrics are architecture and objective dependent, and can therefore not directly be compared"
> > In contrast, this is one of the main strength, we wrote:
> > "However, [resource bounded exploitability] is still useful for comparing different found solution pairs [...] as long as we use the same computational resources to compute approximate best responses against them. Negative values of [resource bounded exploitability] should be interpreted as “robust up to our computational resources to attack it”."

---

> > > ### Author Response · Authors · 2018-11-26
> > > **Thank you and further clarifications**
> > >
> > > Thank you for your interest and suggestions. Please find our comments below:
> > >
> > > - Regarding stochasticity
> > > Here we mean the stochasticity in estimating the performance measure.  It seems like you assume to have a direct access to exact objective values in your paper. However,  in practice, we can only estimate these via samples. In this case, we show that one needs to carefully split the training set in 3 parts in order to maintain an unbiased estimate of the expected duality gap. See an elaborate discussion of this in Section 4 of our manuscript.
> > >
> > > - Regarding your objection to point (6)
> > > We discuss this in Appendix C “Analysis of the quality of the empirical DG”. In particular, see the paragraphs “Collapsed worst case generator ” and “Suboptimal solutions due to the optimization”, where we discuss what happens in the case of mode collapse of the worst case generator and the effect of the number of optimization steps on the quality of the solution.
> > >
> > > We disagree that we are not clear that the theoretical gap is not what is obtained in practice. From our manuscript: “The theoretical assumption appearing in the proof in Appendix A is that the discriminator and generator have unbounded capacity and we can obtain the true minimizer and maximizer when computing u_worst and v_worst, respectively. This, however, is not tractable in practice. Furthermore, it is well known that one common problem in GANs is mode collapse. This raises the question of how the duality gap metric would be affected if the worst generator that we compute is collapsed itself.” Moreover, in Section 4 we dedicate an entire section for the **estimation** of the theoretical gap in practice. Our extensive experiments do show that this estimation is very efficient in evaluating models, yet it is practical to compute.
> > >
> > > - Regarding comparison of different objectives
> > > If one wants to compare two models, e.g. a vanilla GAN and a WGAN, one has to consider that the two objective functions are different. Specifically, the first one contains a log, whereas the latter does not and therefore the range of the values will be different. Furthermore, the landscape of neural networks is still the object of intensive research and we therefore did not want to make any pre-emptive claims without having done a more thorough investigation (which we intend to do as future work) .

---

> > > > ### Public Comment · ~Frans_A_Oliehoek1 · 2018-11-27
> > > > **Probably the last comments needed**
> > > >
> > > > Thanks. I appreciate the continued discussion.
> > > >
> > > > Let me try and wrap up:
> > > >
> > > > -stochasticity: I think there was some confusion about terminology, we also did sample based evaluation (so it was stochastic), but draw new data points (directly from mixture of Gaussian distribution). Indeed, we did not explicitly describe the techniques of splitting in 3 data sets. That is a perfectly valid statement.
> > > >
> > > > -Point 6: My objection was to the statement that our paper does not discuss limitations in applying exploitability ("notable differences are [...]  we discuss how the suboptimality of the solution affects the quality of the score"). But our paper is extremely explicit about this.
> > > >
> > > > I agree that your paper is also explicit on this front in appendix C. I still feel that certain formulations in section 4 are open to misinterpretation. ("we discuss the appropriate way to estimate the metric using samples", "we describe a method for an efficient and practical computation of the duality gap.", "For all experiments we report both the duality gap (DG) and the minimax loss"), but that is for you to consider.
> > > >
> > > > -objective. This is a matter of perspective. I would argue that the designer should pick one true, 'test' objective (say the vanilla GAN objective with log). If WGANs (or any other variant) learns better, it should perform well on this test objective (even if it was trained on another). Exploitability gives a way to compare a GAN and a WGAN model on the test objective.
> > > >
> > > > Again, thank you for the discussion, I find it very interesting work, and I would welcome any later discussions via email or other form it that would be of interest.
> > > >
> > > > Best,
> > > > -Frans

---

### Meta-Review · Area_Chair1 · 2018-12-18
**Revise and resubmit**

**Confidence:** 4
**Recommendation:** Reject

**Metareview:**

All reviewers still argue for rejection for the submitted paper. The AC thinks that this paper should be published at some point, but for now it is a "revise and resubmit".